# Plant-Growth-Promoting Potential of PGPE Isolated from *Dactylis glomerata* L.

**DOI:** 10.3390/microorganisms10040731

**Published:** 2022-03-29

**Authors:** Chaowen Zhang, Kai Cai, Mengyuan Li, Jiaqi Zheng, Yuzhu Han

**Affiliations:** 1College of Animal Science and Technology, Southwest University, Chongqing 402460, China; z1656096288@email.swu.edu.cn (C.Z.); limn0128@email.swu.edu.cn (K.C.); lmy6998181@email.swu.edu.cn (M.L.); zjq2000@email.swu.edu.cn (J.Z.); 2Chongqing Key Laboratory of Herbivore Science, Chongqing 402460, China

**Keywords:** plant-growth-promoting endophytes, *Dactylis glomerata* L., IAA, phosphate solubilization, antimicrobial activity

## Abstract

Plant-growth-promoting endophytes (PGPE) are a kind of beneficial microorganisms which could inhabit plant tissues to antagonize certain plant pathogens and promote the host plant’s growth and development. At present, many studies have confirmed the mutualistic effect of endophytes with plants, but there are few systematic studies on beneficial roles between endophytes and *Dactylis glomerata*, especially on the PGPE characteristics of the forage and environmental restoration plant. This study aimed to isolate PGPE from *D. glomerata*, evaluate their effects on plant growth, and ultimately acquire desirable microbial inoculants for agricultural use. First, endophytes were isolated from *D. glomerata* by plant re-inoculation experiment, and identified by morphological and molecular analyses. Fixation medium and methods were carried out to assess the nitrogen fixation ability of the strains. Then, the ability to dissolve phosphorus was determined by the Olsen and silicate medium methods; secretory IAA was measured by Salkowski colorimetric method; and the inhibitive effects on phytopathogen were observed by confrontation culture. Twenty-one strains were isolated from four varieties of *D. glomerata*, among which 14 strains with plant-growth-promoting characteristics were obtained by re-inoculation experiments, including seven endophytic bacteria and seven endophytic fungi. Further evaluation of three selected strains with the most significant PGP attributes were performed by using the pot re-inoculation experiment which revealed that TG2-B derived from *Myroides odoratimimus* was the most effective plant-growth-promoting agent due to its ability to produce high levels of IAA; the strain *Bacillus cereus* exhibited the most robust ability in dissolving inorganic phosphorus; and *Trichoderma harzianum* manifested a conspicuously antagonistic effect against a variety of plant pathogens. For the first time, this study reports the identification of *D. glomerata* endophytes that were able to promote plant growth and have a considerable antagonistic effects on plant pathogens, which could be considered as microbial inoculants for using in improving crop production and plant disease control.

## 1. Introduction

Chemical pesticides and synthetic fertilizers play an important role in high-yield crops production, but their excessive and long-term use has foreshadowed looming environmental concerns [1]. It is, therefore, highly desirable to develop environment-friendly and sustainable agricultural practices. Some researchers found that eclectic biological approaches can gain these ends. Microbial agents can substitute for agricultural chemicals, some of which have exhibited notable amelioration to the surrounding environments [2], by providing abundant nutrients to crops and improving soil quality and nutrient cycling [3], and ultimately enhancing crops production [4]. As a type of microbial biological agent, plant endophytes seem to be a veritable biological treasure trove. Researchers have isolated 1172 endophytic isolates from *Theobroma cacao* of tropical rainforests [5], manifesting that plant endophytes are valuable agricultural resources. Moreover, plant endophytes show overt growth-promoting effects and antimicrobial/antifungal effects on their host plants [6,7]. There is no doubt that endophytes are valuable ecological resources, and studying the interaction between endophytes and plants may play a significant role in promoting the development of agriculture and maintaining ecological stability.

Endophytes were first defined by Bary in 1866 [8] as microorganisms inhabiting plants [8], which are widespread and able to colonize in healthy plant tissues without causing disease symptoms [9]. Residing within plants, plant-growth-promoting endophytes (PGPEs) can dissolve phosphate and organic phosphorus, as exemplified by plant endophytic *Pseudomonas* isolates that are endowed with a striking phosphorus solubilization ability, effectively providing nitrogen, potassium, and other elements to plants [10]. It is understood that PGPEs produce various plant growth regulators, such as IAA, cytokinin, and gibberellin, that could promote plant growth [11] and provide antimicrobial capability, and increase plant-related antioxidant enzyme activities under environmental adversaries [4]. In addition, *Pseudomonas* isolates could promote Fe^3+^ siderophore production [12], induce plant immune pathways by producing (1-aminocyclopropane-1-carboxylic acid, ACC) aminase that may decrease ethylene production, and alleviate the damages inflicted by various environmental cues [13]. A recent study investigated the prominent antimicrobial function of PGPEs and their underlying mechanisms whereby plant endophytes elicited by pathogenic fungi are able to activate biosynthetic gene clusters and produce chitinase, secondary metabolites phenazine, polyketone compounds, and siderophores as a multifaceted strategy against plant pathogens [14].

*Dactylis glomerata* L. is one of the four major graminaceous forage grasses widely distributed in the world, chiefly in Eurasia and North Africa [15]. Because of its high nutritional value, good palatability, and high sugar content for livestock, *D. glomerata* is regarded as an important fodder crop, providing animal feed, supporting dairy products and meat production in temperate regions [16]. It is also productive and can withstand shade, barrenness, and drought. It has also been used in pasture construction in arid grasslands [17]. Moreover, *D. glomerata* and its associated microorganisms could effectively treat and restore the soils polluted by petroleum products, mercury, and other heavy metals [18,19,20]. 

However, to the best of our knowledge, systematic studies on endophytic bacteria and fungi in *D. glomerata* are still lacking, and the evaluation of endophytic resources in *D. glomerata* from the perspective of disease resistance is long overdue. In this study, the microorganisms residing in *D. glomerata* were screened for growth promotion effect on plants. The PGPEs were evaluated for their capability in phosphorus and potassium solubility, nitrogen fixation, IAA secretion, and antimicrobial properties. Further, selected strains were evaluated for their biocontrol potentials through field rust prevention and control tests. Finally, we report the identification of endophytes that could be used as potential plant antibiotic bioinoculants for biological control.

## 2. Materials and Methods

### 2.1. Plant Sampling

In March 2019, a survey was conducted in Rongchang forage germplasm fields, Southwest University, Chongqing, China (105°17′–105°44′ E, 29°15′–29°41′ N). Different varieties of healthy *Dactylis* seeds were collected, including PI 170344 that came from Turkey, PI 292586 that came from Israel, CF 016352 from Morocco, and “Teyou” from the Beijing Best Grass Industry Co., Ltd. in China. In this area, the average annual rainfall is 1099 mm, the annual accumulative temperature is 6383 °C, the annual photoperiod is 1083 h, the relative humidity is 72%, the average annual temperature is 17 °C, the days exceeding 40 °C are about 15 days (d), the lowest temperature is −4 °C (in January), and the hottest mean temperature is 27 °C (in July). The soil in the experimental site was grey–brown–purple soil and the soil texture was medium loam, with a soil pH of 6.5.

### 2.2. Isolation and Purification of Endophytes

The samples’ leaves, stems, and roots were cut into 1 cm^2^ small pieces by sterile scissors, and the plant tissues were washed by running water for three times prior to being immersed in 75% ethanol for 30 s, sodium hypochlorite 2.5% for 5 min, and followed by thorough rinsing with sterile water [21], placed into potato dextrose agar (PDA) with chloramphenicol (0.1 g/L) (Beijing Luqiao, China) for fungal endophytes and Luria-Bertani (LB) medium (Beijing Luqiao, China) for bacterial endophytes, and cultured at 25 °C for one week. Each set was repeated three times. Isolated microorganisms grown on the plates were purified by dilution separation method as previously described [22].

### 2.3. Pot Re-Inoculation Experiment

We isolated 21 endophytes inoculated into potted *D. glomerata* and found that 14 endophytic bacteria and fungi had better induction characteristics for plants. In order to screen endophytic strains with PGP characteristics, we performed pot re-inoculation experiments. For the formal re-inoculation in *D. glomerata*, the selected endophytic bacteria and fungi were cultured on media in Erlenmeyer flasks, which were placed on an incubator with shaking, at 160 rpm, at 25 °C for 48 h. *D. glomerata* seeds disinfection was performed as described above for plant tissues. The seeds were then sown in pots, with the planting depth around 1 cm. The diameter and height of a pot was 22.5 cm and 16 cm, respectively, filled with about 3 kg soil, and placed in an outdoor environment. The soil characterization and climatic conditions refer to the description in Section 2.1. The plants were irrigated with microbial solutions at a concentration of 10^5−6^ CFU mL^−1^ every 7 d. A range of growth indexes, including leaf length, root length, fresh weight of root, and fresh weight of aboveground parts, were assayed after 60 d. 

### 2.4. Morphological, Physiological Characterization of Endophytes

The 14 strains with plant-growth-promoting characteristics obtained by re-inoculation experiments were further evaluated. After 3–5 days of post-inoculation on LB or PDA medium, the color, size, shape, transparency and uniformity, and other morphology of the colonies of each isolate were measured. The shape of cells were examined as described by Khalifa, Sadia Bashir [23,24].

### 2.5. Molecular Identification of Entophytes

Molecular identification using bacterial 16s rRNA gene and fungal internal transcribed spacer (ITS) rDNA was conducted by PCR and sequence analysis. Each PCR reaction was performed in a 50 μL volume consisting of 2 μL (0.5–10.0 ng) templates DNA, 1 μL upstream and downstream primers, respectively, 25 μL, 2 × PCR Taq Master Mix, 21 μL ultrapure water. The oligo primers designed for universal amplification of a bacterial 16S rRNA fragment were 27f (5′-AGAGTTTGATCCTGGCTCAG-3′) and 1492 r (5′-GGTTACCTTGTTACGACTT-3′). The general primers of the fungi ITS rDNA were ITS1 (5′-TCCGTAGGTGAACCTGCGG-3′) and ITS4 (5′-TCCTCCGCTTATTGATATGC-3′) [25]. The PCR program was carried out on a ProFlex PCR 4484075 thermal cycler (USA). The PCR was conducted at 94 °C for 5 min, followed by 35 cycles of 94 °C for 30 s, 54 °C for 30 s, and 72 °C for 90 s. The amplified products were fractionated by 1% agarose gel electrophoresis and visualized under a UV transilluminator before being sequenced. Multiple sequence alignment was conducted by using the Clustal X 1.8 software package. Phylogenetic trees were generated by the neighbor-joining method using MEGA, and confidence was tested by bootstrap analysis with 1000 repeats. 

### 2.6. Qualitative Detection of Nitrogen Fixation Capacity

The strains were subjected to the test for their potential in nitrogen fixation. The active strains were inoculated onto nitrogen fixation liquid medium (Ashby’s Nitrogen-Free Fluid medium, Shanghai Rebio Co., Ltd., Shanghai, China) at 25 °C for 2–3 d. The strains that had survived on the nitrogen fixation medium were re-inoculated onto a new set of nitrogen fixation media. The experiment was repeated three times. The growth rate of the strains represents their ability in nitrogen fixation [26].

### 2.7. Qualitative Test of Potassium Solubility

The activated strains were inoculated onto silicate medium (Qingdao Hope Bio-Technology Co., Ltd., Qingdao, China) [27] at 25 °C for 2–3 d. When clear zones around the colonies were found, the strains were re-inoculated onto a fresh silicate medium. The persistence of clear zones on silicone medium represents their ability in dissolving potassium.

### 2.8. Quantitative Estimation of Phosphorus Solubilization

Quantitative analysis of phosphorus solubilization was conducted following the approach of Olsen [28]. The Erlenmeyer flasks with the culture of strains on phosphorus medium or in water as negative control were oscillated at 150 rpm at 25 °C for 5 d, and each set was repeated three times. At the end of oscillation, the suspension was centrifuged at 10,000 rpm, and the phosphorus content was measured by Mo–Sb colorimetry [29].

### 2.9. Quantitative Determination of Secretory Auxin

Quantitative tests of IAA production were performed on discolored strains after performing the Salkowski colorimetric method as previously described [30]. Briefly, the culture suspension was centrifuged at 10,000 rpm for 10 min, from which 1 mL supernatant was taken to a new tube containing 50 μL phosphoric acid and 2 mL Salkowski colorimetric solution. The control group contains supernatant only. The test tubes were then placed in the dark for 30 min, prior to measurement of OD530 value. A standard curve of IAA content was calculated according to absorbance, which was used to estimate IAA concentrations in samples [31].

### 2.10. Antimicrobial Activities of Endophytes

Antimicrobial experiments were performed using the confrontation method [20,32] with some modification. In brief, endophytes and pathogen (*Cladosporium tenuissimum*, *Fusarium tricinctum*, *Colletotrichum cliviicola*, *Pantoea agglomerans*, *Nigrospora oryzae*, *Phoma exigua*) were inoculated onto a same plate, nutrient agar–potato dextrose agar (NA–PDA; 1:1) mix plate at 25 °C for 3–5 days. Endophytes were placed at a distance of 25 mm from the pathogens which were inoculated on the center of the plate. Plates without endophytes were used as controls. Antimicrobial rate was measured and recorded by using the following formula after the pathogenic fungi in the control plate covered the plate. The experiment was carried out in three independent replicates.
$$\mathrm{Antimicrobial}\;\mathrm{rate}\left(\%\right)=\frac{\mathrm{Control}\;\mathrm{group}\;\mathrm{diameter}-\mathrm{treatment}\;\mathrm{group}\;\mathrm{diameter}}{\mathrm{Control}\;\mathrm{group}\;\mathrm{diameter}}\times 100\%$$

### 2.11. Effect of the Endophytic Inoculation on Rust Resistance of D. glomerata

The effect of endophyte inoculation on rust resistance of *D. glomerata* was studied in the field experiment. Four strains of endophytic bacteria (TG2-B, CFY1-B, 344G1-B, and 344JI-B) were selected from the previous screening. The seeds were stored in Chongqing Key Laboratory of Herbivore Science, Southwest University. Sterilized seeds were germinated on the filter paper with good moist ventilation and sterilization to germinate; the seedlings thus obtained were soaked in liquid cultures of TG2-B, CFY1-B, 344G2-F, and 344JI-B for two hours, while seedlings treated with sterile water served as controls (CK). The inoculated seedlings were then transplanted into the experimental field, and the root and leaf of *D. glomerata* were irrigated with endophytes solutions with an overdiluted concentration of 10^5^–10^6^ CFU mL^−1^ every 7 days. The experiment was conducted in a randomized block design, and repeated three times, with an area of 1 m^2^ in each plot. In May 2020, the incidence of rust disease in the field was assessed. The five-points sampling method was used in each plot, and 10 strains were checked in each plot to record the incidence. The disease grade assessment of *D. glomerata* was based on the standard of rust investigation records concerning the national standard of the Wheat Stripe Rust Forecast Investigation Standard issued by China in 1995 [33].
$$\mathrm{Incidence}\;\mathrm{rate}=\frac{\mathrm{Number}\;\mathrm{of}\;\mathrm{plants}\;\mathrm{infected}\;\mathrm{with}\;\mathrm{rust}}{\mathrm{Investigate}\;\mathrm{the}\;\mathrm{total}\;\mathrm{strain}}\times 100\%$$
$$\mathrm{Disease}\;\mathrm{index}=\frac{\Sigma \;\left(\mathrm{number}\;\mathrm{of}\;\mathrm{rust}\;\mathrm{plants}\;\mathrm{at}\;\mathrm{all}\;\mathrm{levels}\right)\times \;\mathrm{Disease}\;\mathrm{grade}\;\left(\mathrm{value}\right)}{\mathrm{Total}\;\mathrm{number}\;\mathrm{of}\;\mathrm{plants}\;\mathrm{investigated}\;\times \;\mathrm{Superlative}\;\mathrm{value}}\times 100\%$$
$$\mathrm{Relative}\;\mathrm{protection}\;\mathrm{effects}=\frac{\;\mathrm{Control}\;\mathrm{disease}\;\mathrm{index}\;-\;\mathrm{treatment}\;\mathrm{disease}\;\mathrm{index}}{\;\mathrm{Control}\;\mathrm{disease}\;\mathrm{index}}\times 100\%$$

### 2.12. Statistical Analysis

Experimental data were displayed as average ± standard deviation (SD) values. The significant differences among the processing means were evaluated with one-way analysis of variance (ANOVA) by multiple sample comparison, and variables of replication were followed by the LSD test at *p* < 0.05.

## 3. Results

### 3.1. Pot Re-Inoculation Strains Experiment

To discover the growth-promotion effect of inoculating different bacteria and fungi on plants, we isolated 21 endophytes from *D. glomerata* (Table 1). Among them, seven endophytic bacteria and seven fungi with PGP ability were selected. Leaf length, root length, and fresh weight and dry weight of *D. glomerata* were measured by pot re-inoculation. It was found that endophytic bacteria and fungi had significant growth-promotion effects on *D. glomerata* (Figure 1) as the root length, aboveground part fresh weight, and fresh root of *D. glomerata* were significantly different from the control group (*p* < 0.05). Analysis of variance revealed that the lengths of leaves and roots of *D. glomerata* inoculated with bacteria were overall greater than those inoculated with fungi, but the root length, plant height, and fresh weight of *D. glomerata* inoculated with fungi were significantly higher than those of the control group. The values of root length, plant height, fresh weight of roots, and aboveground parts of *D. glomerata* were the largest in TG2-B, while those of root lengths, leaf lengths, and fresh weights were better in TG2-F and 344G2-F.

### 3.2. Morphological Identification of Endophytic Fungi and Bacteria

Seven *D. glomerata* bacterial endophytes were isolated, purified, and transferred to LB medium and cultured at 25 °C for 3–5 d for morphology observation. As shown in Appendix A, TG2-B was 1.5–2.0 mm in diameter, with its Gram staining being negative. Its colonies appeared shiny, yellowish, and slightly moist, with round and smooth margins. The four bacterial strains, including CFY1-B, TG1-B, TJ1-B, and 586G1-B, were white, opaque, and shiny, round but without smooth edges, showing similar shape in morphology; 344G1-B closely resembled 344J1-B, but was slightly different from the four strains mentioned above, which could probably be *Bacillus* spp.

Seven purified *D. glomerata* endophytic fungi were transferred to PDA medium and cultivated at 25 °C for 3–5 d. As is evident in Appendix A, 344G1-F, 344G2-F, and 344G3-F might be *T. harzianum*, as their colonies were green on the front and white or brown on the back. These cultures showed rapid growth, covering the entire plate over 3–5 d. Spores were green under microscopic examination. TG2-F and TG3-F were presumably *Fusarium* spp., due to their white fluffy colonies, slightly red at the center and crescent-shaped spores. The abaxial surface of 586G1-F and 586G2-F was white and darkened around the fourth day after inoculation, with branched septate hyphae but without discernible spores under microscopy, which could be speculated as *Aspergillus* spp. 

### 3.3. Molecular Identification of Endophytes

The 16s rRNA gene sequences of 344G1-B, 344J1-B, 586G1-B, CFY1-B, TG1-B, TG2-B, and TJ1-B have been deposited in GenBank with accession numbers of OK493783, OK493788, OK493785, OK493786, OK4937854, OK493789, and OK493787 (Table 2, Figure 2). Among them, 344G1-B, 344J1-B, CFY1-B, TG1-B and TJ1-B, and 586G1-B were closely aligned with *B. mycoides* strain SeITE01 (KF280239), *B. pumilus* strain HTI3 (MK521055), *B. cereus* strain IIF4SW (KY218870), *B. pseudomycoides* strain BS20 (KR063200), and *B. cereus* strain ATCC 14579 (AE016877) with 99% identity, respectively. In contrast, TG2-B was more similar to *M. odoratimimus* TH-19 (HF947513). In addition, seven endophytic fungi (344G1-F, 344G2-F, 344G3-F, 586G1-F, 586J1-F, TG2-F, and TG3-F) were clustered into *T. harzianum* (344G1-F, 344G2-F, 344G3-F), *A. fumigatus* (586G1-F, 586J1-F) and *F. proliferatum* (TG2-F, TG3-F) (Table 2, Figure 2). BLAST algorithm showed that 344G1-F (OK445674), 344G2-F (OK445675), and 344G3-F (OK445674) were similar to *T. harzianum* isolate Ctccsj-g-hb40547 (KY750324), with 99% homology, whereas 586G1-F (OK448258) and 586J1-F (OK448259) were highly similar to *A. Fumigatus* (99%); TG2-F (OK448256) and TG3-F (OK448257) were similar to *F. proliferatum* isolates A2S1-D96 (KJ767073).

### 3.4. Qualitative and Quantitative Determination of Nitrogen, Phosphorus, and Potassium in Endophytic Fungi and Bacteria

As shown in Table 3, most of the tested bacteria and fungi were found to have nitrogen fixation abilities. Specifically, the endophytic bacteria 344G1-B, 344G2-B, 586G1-B, CFY-B, and TJ1-B and fungi 344G2-F, 344G3-F, 586G1-F, and TG2-F were able to fix nitrogen. With regards to potassium-dissolving test, significant variations were observed between bacterial endophytes, and relatively weak ability was also observed in fungal strains 344G1-F, 586G1-F, and TG2-F. Phosphorus dissolution amount was tested by molybdenum–antimony anti colorimetry, bringing about diverse results among various endophytes. ANOVA analysis determined that the phosphorus dissolution ability of bacteria was generally better than that of fungi. The highest phosphorus dissolution volume was observed in bacterial strains CFY1-B, TG1-B, TG2-B, and TJ1-B, while 344G3-F and TG2-F performed best among fungi isolates.

### 3.5. Determination of Auxin Production

After activating endophytic fungi, bacterial strains were subjected to Salkowski colorimetry for quantitative analysis of auxin secretion. It was shown that the biosynthesis of IAA in *D. glomerata* was diverse under different treatments of endophytic fungi and bacteria. As shown in Table 3, the secretion of IAA induced by the endophytic bacteria group was significantly higher than that induced by fungi. The endophytic fungi 344G1-F and 344G2-F showed the maximum IAA induction, whereas 586G1-F, 586G2-F, TYG2-F, and TYG3-F were not able to induce IAA.

### 3.6. Antimicrobial Activity of Endophytic Bacteria and Fungi in D. glomerata

To identify the endophytes of *D. glomerata* with the potential capacity in preventing and controlling the prevalence of plant diseases, plate confrontation culture method was used to evaluate antagonistic effects of 14 strains of endophytic bacteria and fungi from *D. glomerata* against six different plant pathogens. It was shown that fungal isolates displayed more remarkable disease-inhibitory effects than bacterial isolates. In particular, 344G2-F exhibited the strongest antimicrobial effect, with rates of 100%, 78%, 100%, 100%, 90.86%, and 78.85% against *P. exigua*, *F. tricinctum*, *C. tenuissimum isolate*, *P. agglomerans*, *N. oryzae,* and *C. cliviicola* strains, respectively (Figure 3).

### 3.7. Field Rust Test

Fungal endophyte 344G2-F displayed the best control effect (Table 4), with rust incidence of 18.60 ± 0.65%, disease index of 8.43 ± 0.05%, and relative protection effects reaching 62.29 ± 0.12%. In order to explore the control of field rust by selecting endophytic bacteria and fungi of *D. glomerata*, a range of parameters, including the incidence, disease index, and relative control effect of rust, were measured in the experimental site during May 2020, coincidental with the most common occurrence of rust. As shown in Table 4, after inoculation with screened endophytic bacteria and fungi, the incidence of rust on *D. glomerata* was less than 30% on average, in contrast to the incidence of the control group at 38.69 ± 0.21%. The disease indexes of inoculation groups were no more than 14%, in contrast to 22.21 ± 0.08% in the control group. Fungal endophyte 344G2-F demonstrated the best effectiveness of rust prevention and control, for its rust incidence was 18.60 ± 0.65%, disease index was 8.43 ± 0.05%, and relative protection effects was 62.29 ± 0.12%.

## 4. Discussion

Plant endophytes are an indispensable resource for the development of microbial agents for biological controls. Our results displayed that 14 out of the 21 strains of endophytic bacteria and fungi isolated from different parts of *D. glomerata* were endowed with growth-promoting effect. In the present study, the dominant residing microorganisms within *D. glomerata* include *B. pseudomycoides*, *B. pumilus*, *B. mycoide*, and *B. cereus*, which fostered stimulation to plant growth. With conspicuous promotion effects on plant growth, *Bacillus* spp. was found in ginseng in abundance [34], of which the major strain was identified as *B. pumilus*. Likewise, *Bacillus* spp. were identified as one of the endophytes of *Elaeagnus* spp. in metagenomic studies [18]. Among other similar studies, an endophytic strain, *B. mycoide,* was identified in *Salicornia bigelovii* [35], and Hassan showed that *B. cereus* isolated from *Teucrium polium* L. leaves was the most advantageous, and identical to CFY1-B, 586G1-B, TJ1-B, TG1-B. Handri isolated over twenty endophytic filamentous fungi from *Vicia faba* L., sesame, soybean rhizosphere, and rhizobia and identified them to be *Fusarium* spp. [36], while Lubna obtained *F. proliferatum* from *alfalfa* [37]. Endophytic *T. harzianum* strains were isolated from *Aloe vera*, but only *T. viride* strain was found in *D. glomerata* by Rozpdek [38]. Therefore, this study represents the first report of endophytic *T. harzianum* in *D. glomerata*. 

Our study aims at screening for potential plant-growth-promoting microbial inoculants, for which a pot re-inoculation experiment was conducted together with the purified microbial agents that were used to test the ability in phosphorus and potassium dissolution, nitrogen fixation, and IAA secretion. Antimicrobial experiments were performed to select for potential plant-growth-promoting inoculants, which were examined by field rust disease resistance tests. Ultimately, TG2-B, CFY1-B, and 344G2-F strains were identified as the potential inoculants for their growth-promoting and antimicrobial activities, among which TG2-B was most effective in inducing plant growth in *D. glomerata*, as its inoculation significantly promoted growth in plant height, root length, root fresh weight, and aboveground part fresh weight by 2×, 2.5×, 6×, and 3×, relative to the uninoculated ones, respectively (*p* < 0.05). The *M. odoratimimus* strains isolated from the rhizosphere were reported to enhance the dry weight, fresh weight, plant height, and other biomass factors in mung bean [39]. Similar observations have been made in wheat, where *M. odoratimimus* strain overtly increased the stem length, root length, fresh weight, and dry weight [40,41]. 

In the antimicrobial experiment, TG2 (*M. odoratimimus*) was found to suppress *Nigrospora oryzae*, but such an effect was not known in *M. odoratimimus*. Instead, *M. odoratimimus* was found to exert a strong inhibitory effect on tobacco scab associated with black star disease. In controlling rust in *D. glomerata*, the disease incidence of the group inoculated with TG2-B was 19.12%, which was significantly lower than 38.69% in the control group (*p* < 0.05). Although the distinction on culture media was not obvious, the field study provided conspicuous evidence. It was speculated that high IAA secretion and phosphorus solubility of TG2-B can comprehensively improve rust immunity, while its metabolites can degrade mycotoxins [42], alleviating the damages inflicted by pests and diseases [32]. 

Our present study demonstrated that CFY1-B had the best phosphorous-solubilization effect. Microbial soluble phosphorus is a promising biofertilizer [32]. Soil phosphorus mainly exists in the form of minerals and cannot be directly absorbed by plants. Some endophytes can convert the soil non-absorbable minerals into bioavailable phosphorus, which can be absorbed and utilized by plants, thereby reducing the use of chemical fertilizers [43]. CFY1-B had a significant growth-promoting effect on *D. glomerata*, second only to TG2-B. In our present work, CFY1-B (*B. cereus*) showed the best phosphorus solubilization effect, with dissolvable phosphorus content of 129.53 mg/L. It also exhibited its ability to produce IAA and nitrogen fixation. *B. cereus* was reported to have a high phosphate-solubilizing index and could secrete IAA [32]. It was reported that an unknown *Bacillus* species showed nitrogen fixing, phosphate-solubilizing, potassium-solubilizing, and significant growth-promotion effect on *D. glomerata* [44]. In the present study, we have clearly demonstrated that *B. cereus* is a potential plant microbial agent that could be used to promote *D. glomerata* growth, as antagonistic experiments have significantly displayed its inhibitory effects on *Fusarium tricinctum* and *Nigrospora oryzae*. The field test showed that the incidence rate of *D. glomerata* rust was 38.69%, and that of CFY1-B inoculum group was 28.43% (*p* < 0.05), which had a significant effect on disease control. The underlying mechanism of antimicrobial activity in *B. cereus* may lie in its production of secondary metabolites, for instance, bacteriocins and lipopeptide antibacterial substances including peptide antibiotics and lysozymes [45]. 

The 344G2-F (*T. harzianum*) not only showed the strongest antimicrobial activity against plant pathogens, but it also significantly promoted plant growth in *D. glomerata*, as was evident by the measurement of plant root length, plant height, root fresh weight, and aboveground fresh weight. Significant increase in soybeans yield was observed upon the inoculation with *T. harzianum*, which was attributed to the strain’s ability in pathogen-inhibitory, IAA production, and phosphorus and potassium solubilization, as well as nitrogen fixation [46]. In the present study, 344G2-F has the ability to secrete IAA, dissolve phosphorus, fix nitrogen, and dissolve potassium. It is therefore likely that 344G2-F functions in a similar manner to *T. harzianum*. 

The antagonism experiments showed that the inhibition rate of endophytic fungi on pathogen was generally better than that of endophytic bacteria. *Magnaporthe* spp. showed the strongest antifungal rate against plant pathogens, among which 344G2-F had significant antibacterial effects against *P. exigua*, *F. tricinctum*, *C. tenuissimum Cooke*, *N. oryzae*, and *C. cliviicola* with high antimicrobial rates of 100%, 78%, 100%, 91%, and 78% (*p* > 0.05), respectively. Previous results have shown that *Trichoderma viride* had 71.76% antifungal rate against *P. exigua*, and *Trichoderma viride* had a significant antifungal effect on *Fusarium* spp.; the inhibition rate of *Fusarium nygama* reached 78.80% [47]. *T. harzianum* showed a significant inhibitory effect on *Fusarium spp.* infecting bananas [48]. *T. harzianum* was effective against *C. graminicola*, showing maximum antifungal effect up to 76.47% [49]. When *T. harzianum* was antagonized against parasitic mango anthracnose, the severity of disease was reduced by 87.90% [50]. Moreover, a few studies have investigated that *C. tenuissimum Cooke* and *N. oryzae* were antagonized by *T. harzianum*. As confirmed by field trial, 344G2-F demonstrated the highest effectiveness in alleviating the rust damage to *D. glomerata*, with the symptomatic incidence of 18.60%, which was significantly lower than the uninoculated group of 38.69% (*p* < 0.05). Our results are well in line with previous studies on wheat, where the incidence of rust disease was substantially lowered as the result of *T. harzianum* inoculation [51]. Overall, our study found that 344G2-F showed a promising level of antagonistic effect on a variety of plant pathogens, indicating that *Trichoderma* spp. could be used as an environment-friendly microbial agent without detrimental effects on the environment.

## 5. Conclusions

Our present study may provide a foundation for the development and utilization of endophytic bacteria and fungi in *D. glomerata*. It has displayed the interaction between the forage plant *D. glomerata* and its endophytes: plants provide a suitable growth environment for endophytes, and endophytes secrete some beneficial substances that promote plant growth. It is revealed that the dominant microorganisms residing in *D. glomerata* are *Bacillus*, *Trichoderma*, *Fusarium*, and *Aspergillus*, as these predominant niches have already been occupied by half of the fungal and bacterial endophytes from *D. glomerata* tissues. These strains contribute to the enhancement of plant growth and crop yield, with the effects of ameliorating plant nutrition, secreting auxin, and inhibiting phytopathogens. *B. cereus*, *T. harzianum*, and *M. odoratimimus* were ultimately selected as excellent microbial inoculants for PGP plant growth promotion and antiphytopathogen, which may replace, at least in part, chemical fertilizers and pesticides over time, and significantly reduce a series of environmental pollutions from production to application of chemical fertilizers and pesticides.

## Figures and Tables

**Figure 1 microorganisms-10-00731-f001:**
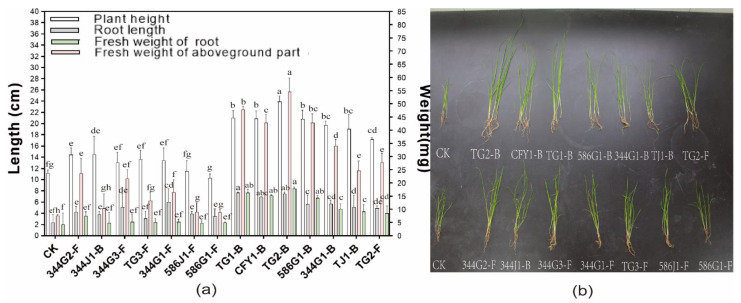
(**a**) Effects of endophytic fungi and bacteria treatment on plant biomass of *D. glomerata*, different letters on bars mean significantly different at *p* < 0.05. (**b**) Seedlings showing differences in root and shoot lengths between control and endophytic fungi and bacteria treatments on *D. glomerata* seedlings after 60 days grown in pots.

**Figure 2 microorganisms-10-00731-f002:**
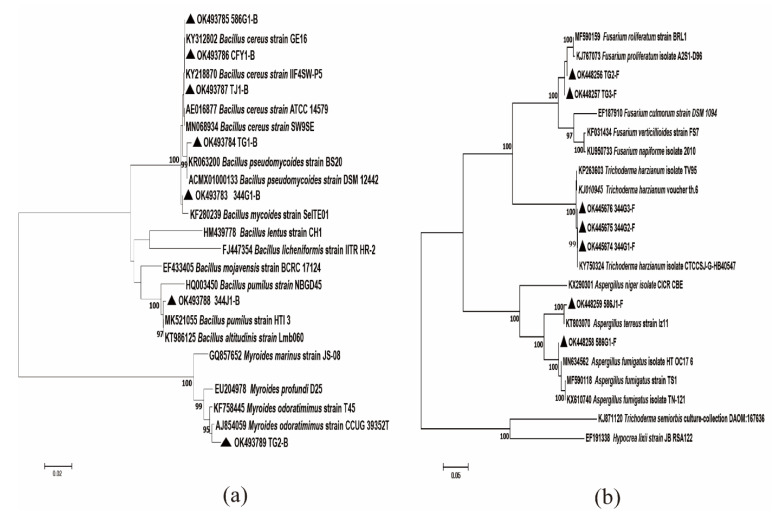
Neighbor-joining tree deduced through MEGA 7.0 software based on the sequences of 16S and ITS rRNA gene of endophytes; (**a**) is endophytic bacteria and (**b**) is endophytic fungi.

**Figure 3 microorganisms-10-00731-f003:**
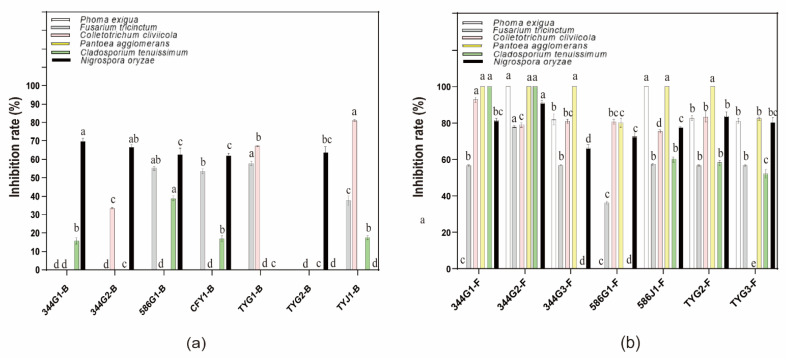
Inhibition rate of endophytic fungi and bacteria: (**a**) is endophytic bacteria of inhibition rate; (**b**) is endophytic fungi of inhibition rate. Different letters on bars mean significantly different at *p* < 0.05.

**Table 1 microorganisms-10-00731-t001:** Endophytes isolated from different varieties and different parts of *Dactylis glomerata*.

*Dactylis glomerata* Sample Name	Isolated Strains (Strain Isolation Site)	
	Bacteria	Fungi
PI170344	344J1-B(stem)	344G1-F(root)
	344G1-B(root)	344G2-F(root)
		344G3-F(root)
		344G4-F(root)
		344Y1-F(leaf)
		344Y2-F(leaf)
		344Y3-F(leaf)
PI292586	586Y1-B(leaf)	586G1-F(root)
	586G1-B(root)	586J1-F(stem)
		586J2-F(stem)
CF016352	CFY1-B(leaves)	
Teyou	TG1-B(root)	TG1-F(root)
	TG2-B(root)	TG2-F(root)
	TJ1-B(stem)	TG3-F(root)
Total	8	13

Notes: “()” represents endophytes isolated from different parts of *Dactylis glomerata*.

**Table 2 microorganisms-10-00731-t002:** The molecular identification of endophytic microbial isolates from *D. glomerata*.

	Microbial Isolates	Nearest Homolog Sequences (Accession Number)	Sequences Identity (%)	Accession Number
Bacterial endophytes	344G1-B	*Bacillus mycoides strain* SeITE01 (KF280239)	99	OK493783
	344J1-B	*Bacillus pumilus strain* HTI3(MK521055)	99	OK493788
	586G1-B	*Bacillus cereus strain* GE16(KY312802)	98	OK493785
	CFY1-B	*Bacillus cereus strain* IIF4SW (KY218870)	99	OK493786
	TG1-B	*Bacillus pseudomycoides strain BS20 (KR063200)*	99	OK493784
	TG2-B	*Myroides odoratimimus strain T45 (KF758445)*	99	OK493789
	TJ1-B	*Bacillus cereus strain ATCC 14579 (AE016877)*	99	OK493787
Fungal endophytes	344G1-F	*Trichoderma harzianum isolate CTCCSJ-G-HB40547 (KY750324)*	93	OK445674
	344G2-F	*Trichoderma harzianum isolate CTCCSJ-G-HB40547 (KY750324)*	93	OK445675
	344G3-F	*Trichoderma harzianum isolate CTCCSJ-G-HB40547 (KY750324)*	93	OK445676
	586G1-F	*Aspergillus fumigatus isolate HT OC176(MN634562)*	100	OK448258
	586J1-F	*Aspergillus fumigatus isolate HT OC176(MN634562)*	100	OK448259
	TG2-F	*Fusarium proliferatum isolate A2S1-D96 (KJ767073)*	98	OK448256
	TG3-F	*Fusarium proliferatum isolate A2S1-D96 (KJ767073)*	98	OK448257

**Table 3 microorganisms-10-00731-t003:** Test of phosphorus, potassium, and nitrogen fixation and auxin production ability of endophytic bacteria and fungi.

Microbial Isolates	Phosphorus Quantitative Test (mg/L)	Nitrogen Fixation Capacity	Potassium Dissolving Capacity	IAA (μg/m)
344G1-B	60.52 ± 0.50 ^g^	+	+	10.21 ± 0.98 ^b^
344J1-B	20.54 ± 0.21 ^k^	+	+	6.11 ± 0.53 ^c^
586G1-B	70.97 ± 0.44 ^f^	+	+	5.38 ± 0.35 ^c^
CFY1-B	129.53 ± 0.35 ^a^	+	+	5.73 ± 0.44 ^c^
TG1-B	121.55 ± 2.6 ^c^	-	+	4.55 ± 0.65 ^cd^
TG2-B	90.15 ± 0.02 ^e^	-	+	52.57 ± 3.73 ^a^
TJ1-B	126.36 ± 0.35 ^b^	+	+	2.43 ± 0.58 ^d^
344G1-F	23.83 ± 0.15 ^j^	-	-	5.44 ± 1.20 ^b^
344G2-F	32.82 ± 0.60 ^h^	+	+	10.46 ± 2.40 ^a^
344G3-F	29.70 ± 0.22 ^i^	+	+	4.87 ± 0.67 ^c^
586G1-F	11.09 ± 1.06 ^m^	+	-	0 ^e^
586J1-F	13.44 ± 1.97 ^l^	-	+	0 ^e^
TYG2-F	29.65 ± 0.10 ^i^	-	-	0 ^e^
TYG3-F	21.61 ± 0.07 ^k^	+	+	0 ^e^

Notes: Values within the same column with different letters are significantly different (*p* ≤ 0.05) by LSD test, values are means ± SE (n = 6). -,+, denotes no, and have nitrogen fixation capacity and potassium dissolving capacity, respectively.

**Table 4 microorganisms-10-00731-t004:** Incidence of experimental rust in 2020.

Microbial Isolates	Incidence %	Disease Index %	Relative Protection Effects %
344J1-B	25.47 ± 0.57 ^c^	13.60 ± 0.04 ^b^	38.74 ± 0.03 ^d^
CFY1-B	28.43 ± 0.58 ^b^	13.22 ± 0.25 ^c^	41.44 ± 0.03 ^c^
TG2-B	19.12 ± 0.57 ^d^	8.79 ± 0.09 ^d^	60.56 ± 0.38 ^b^
344G2-F	18.60 ± 0.65 ^d^	8.43 ± 0.05 ^e^	62.29 ± 0.12 ^a^
CK	38.69 ± 0.21 ^a^	22.21 ± 0.08 ^a^	0.00 ^e^

Notes: Values within the same column with different letters are significantly different (*p* ≤ 0.05) by LSD test, values are means ± SE (n = 3). *Dactylis glomerata* seedlings were inoculated separately: 344J1-B, CFY-B, TG2-B, 344G2-F. Seedlings treated with sterile water served as controls (CK).

## Data Availability

Not applicable.

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
