# Peer review of "Plant-Growth-Promoting Potential of PGPE Isolated from *Dactylis glomerata* L."

_microorganisms, 2022, doi:10.3390/microorganisms10040731_

Round 1
Reviewer 1 Report
Article is about isolation and characterization of endophytes from Dactylis glomerata plant. From 21 isolated strains of bacteria and fungi were based on pot re-inoculation experiment selected 7 bacterial and 7 fungal endophytes for testing PGP characteristics.
Problem are not explained thinks like pot re-inoculation experiment where is written selected endophytic bacteria and fungi… but for me is there missing information based on what were selected?
I would never conclude that in Dactylis glomerate Bacillus spp. were found as the dominant residing microorganisms based on identification of 7 bacteria which were selected from bigger set.
Strange is that for sterilization of plant material (plant, seeds) different procedures are used.
For isolation of bacteria was used LB and for fungi PDA. On PDA grow also bacteria, logical would be use of selective media for fungi.
In text is lots of small mistakes in organisms names (non italic, capital letters).
In PCR reaction is written volume of template DNA but not concentration (line 127).
In table 3 there is no unit for Phosphorus quantitative test. Why are IAA, phosphorus and nitrogen in two tables and not in one. In IAA unit µl/m.
In antimicrobial effect I would appreciate if selection of P. exigua, F. tricinctum, C. tenuissimum, P. agglomerans, N. oryzae and C. cliviicola would be explained.
Text in lines 45-48 need revision
Sentence in lines 49-51 need to be rewritten.
Line 79 anti-amicrobial instead anti-microbial
Lines 85-89 sentence need revision
Line 100 funal
Line 102 culture
Line 111 by with
Line 117 overground
Line 229-233 whole paragraph should not be in text.
Line 243-247 whole paragraph should not be in text.
Line 321 isolate
Table 5 not explained what is CK.
Line 382 – 386
Subject of article could be interesting but problem is that text is not well written e.g. “Such an effect is likely due to the high IAA secretion and phosphorus solubility of TG2-B, which can comprehensively improve rust immunity, while its metabolites can degrade mycotoxins…”. Therefore whole text need corrections and improving to be more understandable for readers.
Author Response
Dear reviewer 1,
We appreciate your rigorous attitude, professional evaluation and constructive suggestions. Thank you so much for spending a lot of valuable time to review our paper. We have solved the problems you raised one by one, all our changes of manuscript are displayed in “Trace Change “function of MS Word. With best wishes for happiness in your life and work.
With kindest regards,
Dr. Yuzhu Han
1.Problem are not explained thinks like pot re-inoculation experiment where is written selected endophytic bacteria and fungi… but for me is there missing information based on what were selected?
Reply: This information has been added and elaborated in revised manuscript. We isolated 21 endophytes from D. glomerata (Table 1). Among them, seven endophytic bacteria and seven fungi with PGP ability were selected to further verify their promoting effects.
2.I would never conclude that in Dactylis glomerate Bacillus spp. were found as the dominant residing microorganisms based on identification of 7 bacteria which were selected from bigger set.
Reply: I agree with you. We have modified to the specific Bacillus species.
3.Strange is that for sterilization of plant material (plant, seeds) different procedures are used.
Reply: We are grateful of your meticulous to point out this problem. The disinfection procedures of seed and tissue are the same. We missed some information in previous article. Now we have modified in revised paper.
The plant tissues were washed by running water for three times prior to being immersed in 75% ethanol for 30 seconds, sodium hypochlorite 2.5% for 5 min, and followed by thorough rinsing with sterile water,D. glomerata seeds disinfection was performed as described above for plant tissues.
We refer to the disinfection method of Ref (1) and made some modification.
Reference:
1.Hassan, S.E.-D. Plant Growth-Promoting Activities for Bacterial and Fungal Endophytes Isolated from Medicinal Plant of Teucrium Polium L. Journal of Advanced Research 2017, 8, 687–695, doi:10.1016/j.jare.2017.09.001.
4.For isolation of bacteria was used LB and for fungi PDA. On PDA grow also bacteria, logical would be use of selective media for fungi.
Reply: We appreciate you pointing out this problem, and we have revised it.
The PDA used in this study had been supplemented with chloramphenicol (0.1 g/L) for screening fungal isolates.
5.In text is lots of small mistakes in organisms names (non italic, capital letters).
Reply: We have carefully revised these mistakes in revised manuscript.
6.In PCR reaction is written volume of template DNA but not concentration (line 127).
Reply: Thanks to the reviewers for their careful discovery , the concentration of template DNA 2µL (0.5–10.0 ng) has been supplemented.
1.Khan, M.S.; Gao, J.; Zhang, M.; Chen, X.; Zhang, X. Isolation and Characterization of Plant Growth-Promoting Endophytic Bacteria Bacillus Stratosphericus LW-03 from Lilium Wardii. 3 Biotech 2020, 10, doi:10.1007/s13205-020-02294-2.
7.In table 3 there is no unit for Phosphorus quantitative test. Why are IAA, phosphorus and nitrogen in two tables and not in one. In IAA unit µl/m.
Re: We appreciate you pointing out this problem, we have revised the table.
8.In antimicrobial effect I would appreciate if selection of P. exigua, F. tricinctum, C. tenuissimum, P. agglomerans, N. oryzae and C. cliviicola would be explained.
Reply: Re: This six microbial strains are the most common plant pathogens in the world, especially in Chongqing, China. All of them associated with crop failure, leading to great economic losses,for example, P. exigua causes necrosis of potato tubers[1]; F. tricinctum causes disease in corn and the secondary metabolism of the Fusarium seriously leads to food safety and human health[2]; C. tenuissimum can cause mango trees to be severely affected [3]; Date palm diseases of date palm are caused by N. oryzae, while causing great economic losses[4]; C. cliviicola has been reported to cause leaf spot disease in tobacco[5] .
Reference:
- Bain, R.A.; Wastie, J. Variation in Pathogenicity among Isolates Of Phoma Exigua Var.Foveata on Potato Cultivars Differing in Resistance. Potato Research 1982, doi:10.1007/BF02357279.
- Cuomo, V.; Randazzo, A.; Meca, G.; Moretti, A.; Cascone, A.; Eriksson, O.; Novellino, E.; Ritieni, A. Production of Enniatins by Fusarium Tricinctum in Solid Corn Culture: Structural Analysis and Effects on Mitochondrial Respiration. Food Chemistry 2013, 140, 784–793, doi:10.1016/j.foodchem.2012.10.136.
- Guillén-Sánchez, D.; Yañez-Morales, M.; Teliz, D.; Siebe, C.; Baños, S. Morphological and Molecular Characterization of Cladosporium Tenuissimum Cooke (Deuteromycotina: Hyphomycetes) on Mango Tree Panicles: Symptoms, Pathogenicity and Severity of the Fungus. http://dx.doi.org/10.1051/fruits:2007032 2007, 62, doi:10.1051/fruits:2007032.
- Abass, M.H.; Mohammed, N.H. Morphological, Molecular and Pathological Study on Nigrospora Oryzae and Nigrospora Sphaerica, the Leaf Spot Fungi of Date Palm. 2014. doi(PDF) Morphological, molecular and pathological study on Nigrospora oryzae and Nigrospora sphaerica, the leaf spot fungi of date palm (researchgate.net)
- Wang, Y.R.; Hu, Z.; Zhong, J.; Chen, Y.; Zhu, J.Z. First Report of Colletotrichum Cliviicola Causing Leaf Spot on Tobacco (Nicotiana Tabacum) in Hunan Province of China. Plant Disease 2021, doi:10.1094/PDIS-02-21-0409-PDN.
9.Text in lines 45-48 need revision
Sentence in lines 49-51 need to be rewritten.
Line 79 anti-amicrobial instead anti-microbial
Lines 85-89 sentence need revision
Line 100 funal
Line 102 culture
Line 111 by with
Line 117 overground
Line 229-233 whole paragraph should not be in text.
Line 243-247 whole paragraph should not be in text.
Line 321 isolate
Table 5 not explained what is CK.
Reply: Thank you for patiently pointing out these problems. We proofread and correct the above problems one by one, CK: Seedlings treated with sterile water served as controls (CK), the explanation of CK has added in Table 5.
Reply: Thank you for patiently pointing out these problems. We proofread and correct the above problems one by one, CK: Seedlings treated with sterile water served as controls (CK), the explanation of CK has added in Table 5.
10.Line 382 – 386, Subject of article could be interesting but problem is that text is not well written e.g. “Such an effect is likely due to the high IAA secretion and phosphorus solubility of TG2-B, which can comprehensively improve rust immunity, while its metabolites can degrade mycotoxins…”Therefore whole text need corrections and improving to be more understandable for readers.
Reply: Thanks for your constructive reminders. We have made careful modifications to this part and the whole text.

Reviewer 2 Report
the manuscript "Plant growth promoting potential of PGPE isolated from Dactylis glomerata L." is a very well presnted manuscript
I have only two minor comments
The quality of figure 2 is to low
and the Ref (17) many datails missing (year doi etc)
I suggest to accept the manuscript
Author Response
Dear reveiwer 2 ,
Thank you for your recognition of our work,kind comments and helpful suggestions , we have solved the problem you raised one by one, all our changes of manuscript are displayed in "Trace Change " function of MS Word . May happiness and health be with you always.
Yours Sincerely
Dr. Yuzhu Han
1.The quality of figure 2 is to low
Reply: Thank you so much for your constructive suggestions. Figure 2 has been replaced with a higher quality and clearer picture.
2.the Ref (17) many datails missing (year doi etc)
Reply: The Ref (17) missing information has been added. However, this reference has no DOI. We supply the website where this article is located instead.
Ref (17): Farshadfar, M. Evaluation of the Forage Yield and Quality in Some Accessions of Dactylis Glomerata under Irrigated Conditions. Scholars Research Library 2012, 3 (4), 6,
doi:https://www.scholarsresearchlibrary.com/articles/evaluation-of-the-forage-yield-and-quality-in-some-accessions-of-dactylis-glomerata-under-irrigated-conditions.pdf
